# Cobalt-free composite-structured cathodes with lithium-stoichiometry control for sustainable lithium-ion batteries

Ke Chen [1], Pallab Barai[2], Ozgenur Kahvecioglu [2], Lijun Wu [1], Krzysztof Z. Pupek[2], Mingyuan Ge [1], Lu Ma[1], Steven N. Ehrlich[1], Hui Zhong[3], Yimei Zhu [1], Venkat Srinivasan[2], Jianming Bai [1]✉ & Feng Wang [1,2]✉

Lithium-ion batteries play a crucial role in decarbonizing transportation and power grids, but their reliance on high-cost, earth-scarce cobalt in the commonly employed high-energy layered Li(NiMnCo)O$_2$ cathodes raises supply-chain and sustainability concerns. Despite numerous attempts to address this challenge, eliminating Co from Li(NiMnCo)O$_2$ remains elusive, as doing so detrimentally affects its layering and cycling stability. Here, we report on the rational stoichiometry control in synthesizing Li-deficient composite-structured LiNi$_{0.95}$Mn$_{0.05}$O$_2$, comprising intergrown layered and rocksalt phases, which outperforms traditional layered counterparts. Through multiscale-correlated experimental characterization and computational modeling on the calcination process, we unveil the role of Li-deficiency in suppressing the rocksalt-to-layered phase transformation and crystal growth, leading to small-sized composites with the desired low anisotropic lattice expansion/contraction during charging and discharging. As a consequence, Li-deficient LiNi$_{0.95}$Mn$_{0.05}$O$_2$ delivers 90% first-cycle Coulombic efficiency, 90% capacity retention, and *close-to-zero* voltage fade for 100 deep cycles, showing its potential as a Co-free cathode for sustainable Li-ion batteries.

Nickel-manganese-cobalt (NMC) based cathode active materials (CAMs) with high Ni content are preferred in lithium-ion batteries (LIBs), especially for those powering electric vehicles, due to their high storage capacity and low cost[1,2]. However, these high-Ni CAMs face critical issues related to surface reconstruction, oxygen release, transition metal (TM) dissolution, bulk fatigue, and cracking, precluding their practical applications[3–7]. Over the past decade, significant efforts have been invested to alleviate these issues, primarily focusing on LiNi$_{1-x-y}$Mn$_x$Co$_y$O$_2$ (1-x-y ≥ 0.8) with a layered structure[8,9]. Co plays a crucial role in promoting structural ordering and cycling stability of NMC through its facilitation of Li/Ni ordering during calcination[10–12]. However, the increasing cost, environmental pollution from Co mining, and supply shortage call for eliminating Co from LIBs[13–17].

As illustrated in Fig. 1a, Ni/Mn-based Co-free cathodes, (LiNi$_{1-x}$Mn$_x$O$_2$), are attractive for their high capacity, high thermal stability, and safety. However, they suffer from Li/Ni disordering and cycling instability due to the introduction of Ni$^{2+}$ to maintain charge neutrality[18–21]. Growing evidence shows that Li/Ni mixing is unavoidable in the Ni/Mn-based CAMs and becomes worse at higher Mn content, resulting from the increased interplane super-exchange and intraplane magnetic frustration among the magnetic Ni$^{2+}$, Ni$^{3+}$, and Mn$^{4+}$ ions[22–28]. Consequently, structural disordering induced capacity decay and voltage fade have been the major challenges for the practical use of Co-free layered cathodes, despite numerous efforts over the past decade[16,17,22].

High-Ni NMC CAMs are generally synthesized from their hydroxide counterparts through high-temperature calcination with Li source

[1]Brookhaven National Laboratory, Upton, NY 11973, USA. [2]Argonne National Laboratory, Lemont, IL 60439, USA. [3]Department of Joint Photon Science Institute, Stony Brook University, Stony Brook, NY 11794, USA. ✉e-mail: jmbai@bnl.gov; fengwang@anl.gov

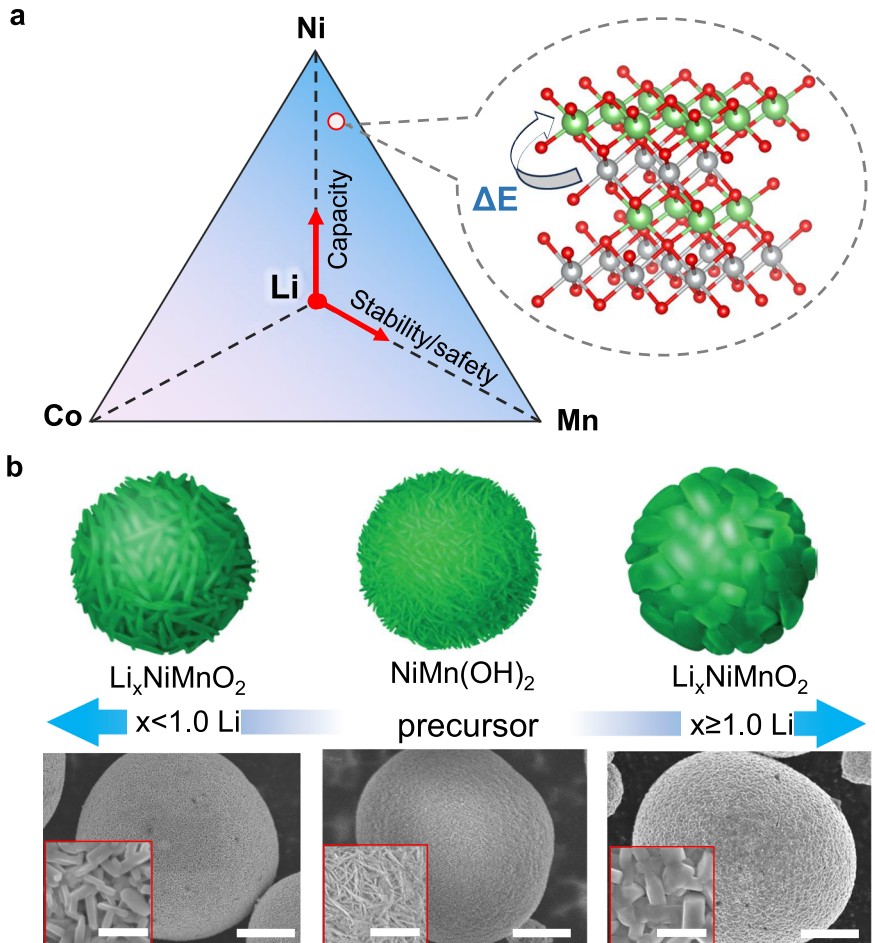

**Fig. 1 | Li stoichiometry as a tuning handle for the structural and morphological control during synthesis of Ni/Mn-based CAMs. a** Composition-property correlation in the Li-Ni-Mn-(Co) space, highlighting the performance merits in terms of capacity, cycling stability and safety, which, however, may be compromised by the unavoidable Li/Ni mixing (inset), arising from the low energy barrier ($\Delta E$) for Ni migration to Li sites within octahedra of the layered framework in the absence of Co (Red: Oxygen, Green: Li, Gray: Ni/Mn). **b** Morphological tuning via Li stoichiometry, demonstrated by the abrupt difference in particle size and shape of calcined $Li_xNi_{0.95}Mn_{0.05}O_2$ at $x = 0.95$ (left) and $x = 1.05$ (right) from the same hydroxide precursor ($Ni_{0.95}Mn_{0.05}(OH)_2$; middle). Top: cartoons; Bottom: scanning electron microscopy (SEM) images. Scale bars: 5 μm in Fig. 1b and 1 μm in the insets, respectively.

in an oxygen atmosphere. The calcination process involves multiphase transformation and strongly depends on the Li and TM composition. It is now becoming known that lithiation, with the simultaneous incorporation of Li and O into the crystal lattice, is crucial to the phase transformation and crystallization processes. However, it remains unclear how the calcining conditions cause the vastly varying particle size, ranging from tens to hundreds of nanometers (as illustrated in Fig. 1b). In the common practice of synthesizing high-Ni CAMs, excess Li source (e.g., lithium hydroxide, LiOH) is added to the precursors to compensate for Li loss, ensuring the formation of Li-stoichiometric layer-structured CAMs in the calcined product[29–31]. However, the added extra Li causes formation of large-sized cuboid-shaped particles, with diameters up to a few hundred nanometers (Fig. 1b; right), which is in stark contrast to the small, tens of nanometer-sized rod-like particles in Li-deficient CAMs (Fig. 1b; left). On the other hand, extra Li turns into resistive Li residual on the particle surface, which induces surface instability during cycling[32,33].

Despite many reports on the synthesis of Li-excess and Li-stoichiometric CAMs, very few studies have explored the Li-deficient regime (with the lithium over transition metal ratio, Li/TM less than 1) to elucidate the role of Li stoichiometry in tuning their structural and morphological properties[34,35]. In a recent report, composite-structured LiNiO2 (LNO) consisting of rock salt (RS) and layered phases was synthesized by controlling Li incorporation during calcination,

resulting in improved cycling stability compared to the traditional layered LNO[36]. Herein, we report rational control over Li-stoichiometry in synthesizing Co-free $Li_xNi_{0.95}Mn_{0.05}O_2$ (NM9505) from hydroxide precursors, with the nominal Li content ($x$) varying in a wide range, from 0.9 to 1.1. We demonstrate the strong dependence of the structure and morphology of the calcined NM9505 on Li-stoichiometry. As $x$ decreases, the synthesized NM9505 CAMs transition from large layer-structured particles to small-sized composites, consisting of the major layered phase and minor RS epitaxially intergrown within the same cubic-closest packed ($ccp$) oxygen framework. With the decrease in Li-stoichiometry ($x$), both the primary particle size and the size of layered crystallites in NM9505 decrease.

Further process analysis, using in situ synchrotron X-ray diffraction (XRD) correlated with multiscale modeling, reveals the crucial role of lithiation in driving structural ordering and crystal growth during the calcination of NM9505. With excess Li, liquid-phase sintering occurs, leading to large primary particles as commonly reported in the literature[29,30]. With deficient Li, small-sized particles form, consisting of intergrown RS and layered phases. In contrast to the fast degradation of Li-excess layered NM9505, the composite NM9505 with 5% Li deficiency exhibits excellent structural stability, offering 90% first-cycle Coulombic efficiency (CE), *close-to-zero* voltage fade, and 90% capacity retention for 100 deep cycles up to 4.4 V. These findings demonstrate a new route to stabilizing Co-free cathodes through Li-stoichiometry

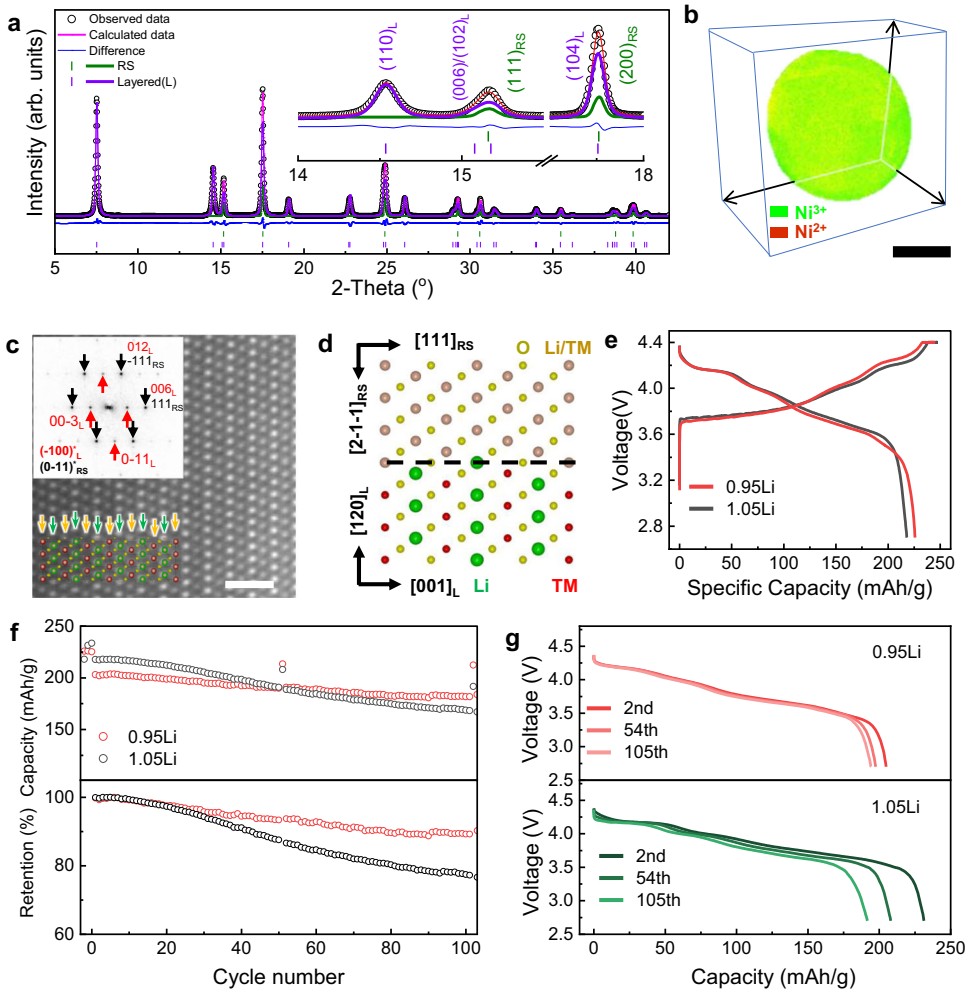

**Fig. 2 | Structural and electrochemical properties of NM9505 with Li deficiency (0.95 Li) and Li excess (1.05 Li). a** Synchrotron XRD pattern in comparison to the calculated one via Rietveld refinement for NM9505-0.95Li ($\lambda = 0.6199$ Å). The XRD pattern was refined using a two-phase model, with a hexagonal layered structure (space group $R\bar{3}m$) and Li-containing RS (space group $Fm\bar{3}m$). See Fig. S1 for the synchrotron XRD pattern and calculated one via Rietveld refinement for NM9505-1.05Li. Insets: Zoom-in patterns at 14°–18° show the individual components, RS and layered (L), within the composite structure. **b** 3D-TXM map of the Ni valence distribution in NM9505-0.95Li (green represents $Ni^{3+}$ and red represents $Ni^{2+}$, while the uniform yellowish color shows a mixture of $Ni^{3+}$ and $Ni^{2+}$). Scale bar: 5 μm. **c** High-resolution HAADF-STEM image taken from the local region of a primary particle, showing a mixture of RS and layered structure. The [0-11] projection of the RS and

[100] projection of the layered structure are embedded in the images. Orange, yellow, green, and red spheres represent Li/TM, O, Li and TM, respectively. FFTs are indexed based on RS $(0-11)^*_{RS}$ and layered structure $(100)^*_L$, respectively. Subscripts RS and L represent RS and layered structure, respectively. See Fig. S4 and Supplementary Note 3 for more HAADF-STEM images for surveying different regions. Scale bar: 1 nm. **d** Illustration of the intergrown RS and layered structure, sharing the same *ccp* oxygen framework. Colors: Orange, yellow, green and red spheres represent Li/TM, O, Li and TM, respectively. **e** The first cycle charge/discharge profile of NM9505-0.95Li and NM9505-1.05Li. **f** The specific and normalized capacity of NM9505-0.95Li and 1.05Li at 0.5 C, with the capacity retention of 90% and 75% after 100 cycles. **g** The 0.1 C discharge profile of NM9505-0.95Li and NM9505-1.05Li at the 2nd, 54th, and 105th cycles.

control, with the desired cost-efficiency by eliminating the extra coating/doping steps[37,38]. With insights into the process-structure-property correlation, this work offers new opportunities to develop high-performance Co-free cathodes for commercially viable and sustainable LIBs.

## Results
### Synthesis of composite-structured NM9505 with Li-stoichiometry control
Following the common practice of adding extra Li during the calcination for synthesizing stoichiometric Ni-based layered oxides, we added 5% extra Li in making NM9505 (NM9505-1.05Li) to compensate for Li loss during calcination. As expected, a highly ordered hexagonal layered structure (space group $R\bar{3}m$) with low Li/Ni mixing (2.6%) was obtained in the synthesized NM9505-1.05Li (Fig. S1 in the supplementary information, SI). By reducing the amount of LiOH·$H_2O$ to Li/

TM = 0.95, we obtained the NM9505-0.95Li with a composite structure consisting of a major layered phase and a minor Li-containing RS phase ($Li_yTM_{1-y}O$, space group $m\bar{3}m$), of about 19.6 mol%, determined by Rietveld refinement of the synchrotron XRD data.

As shown by the zoom-in view in Fig. 2a, the XRD pattern taken from NM9505-0.95Li was well fit using the two-phase model. With a direct comparison between the calculated and experimental curves between 14° and 18°, the peak at ~14.5° can be assigned to the (110) of the layered phase and peaks at angles of 15.2° and 17.5° are associated with both RS and layered phases. Without adding the RS phase, the fitting to these peaks is significantly inaccurate (Fig. S2 and Supplementary Note 1). During the synthesis of CAMs, Li-containing RS formed as an intermediate phase, coexisting with the layered phase for a long period during the slow 2-phase transformation process[39,40]. When deficient Li was provided, a small amount of RS was retained.

The local distribution of the two phases (layered and RS) and their intergrowth within individual secondary and primary particles were further revealed by a combination of three-dimensional (3D) transmission X-ray microscopy (TXM)/X-ray absorption near-edge structure (XANES) spectroscopy and high-angle annular dark-field (HAADF) scanning transmission electron microscopy (STEM)[41]. As shown by the 3D reconstruction of the TXM-XANES map from one secondary particle of NM9505-0.95Li (Fig. 2b), $Ni^{2+}$ is homogenously distributed in the whole particle without segregation, indicating the high mixing of the layered and RS phases. More details are provided in the SI (Supplementary Movie 1-3, Fig. S3, Supplementary Note 2). The local distribution of the layered and RS phases within individual primary particles was further revealed by atomic HAADF-STEM images, as shown in Figs. 2c and S4. In the Z-contrast HAADF-STEM image (with intensity approximately proportional to $Z^{1.7}$, where Z is an atomic number), Li and O atoms are invisible due to their low atomic number. But the layered arrangement, with weak and strong contrast arising from TMs (Ni, Mn), can be clearly seen, which indicates the intergrowth of RS and layered structure (overlapping along beam direction). As in the simulated atomic arrangement (bottom inset), the layers with strong contrast, indicated by green arrows, are the overlap of the TM layers (red spheres) in the layered structure with the Li/TM (orange) in RS. On the other hand, those with weak contrast, indicated by orange arrows, show the overlap of Li layers (green) in the layered structure with Li/TM (orange) in RS. In the fast Fourier transformation (FFT; top inset in Fig. 2c) from this area, the spots indicated by the black arrows represent contributions from both layered structure and RS, thus showing stronger intensity than those indicated by the red arrows, which exclusively belong to the layered structure. More images obtained by sampling different areas (provided in Fig. S4) showed the local RS-rich and layered phase-rich regions, but they are highly mixed at the nanometer scale (as explained in Supplementary Note 3).

By a combination of the bulk XRD study with the local TXM and STEM analysis at different length scales, we were able to confirm the formation of the nanocomposites in NM9505 with 0.95Li, consisting of the main layered phase and minor RS-structured phase intergrown within the same *ccp* oxygen framework, with the epitaxial orientation relationship (illustrated in Fig. 2d).

To interrogate the structure-property correlation, we measured the electrochemical performance of the NM9505 with 0.95Li and 1.05Li at a voltage range of 2.7–4.4 V (Fig. 2e). The two display an overall similar voltage profile. The slightly higher overpotential at the end of charge and discharge in NM9505-0.95Li was due to the presence of RS with low Li diffusivity, which was also shown by the shift of the H2 to H3 transition peaks in the dQ/dV curve in Figure S5. NM9505-0.95Li delivered high first-cycle Coulombic efficiency (CE > 90%) and high discharge capacity (226 mAh/g), much higher than NM9505-1.05Li (85% and 218 mAh/g, respectively). The dQ/dV curves were derived from the charge/discharge profiles, showing the redox peaks associated with the H1, M, H2, and H3 phase transitions (Fig. S5). Among them, the H2 to H3 transition at ~4.2 V is considered one of the major issues causing cycling instability in high-Ni cathodes, which is generally explained by the structural collapse and cracks in the cathode particles induced by the large volume change[42]. NM9505-0.95Li showed less intense and broader redox peaks, particularly in the one associated with H2-H3 transition, compared to NM9505-1.05Li. This indicates a smoother phase transition in NM9505-0.95Li compared to NM9505-1.05Li due to the decrease in anisotropic structural change in the composite structure[36,43]. As expected, NM9505-0.95Li maintained high-capacity retention of 90% after 100 cycles at a 0.5 C rate, significantly outperforming NM9505-1.05Li (75%; Fig. 2f). Most of the capacity was recovered when the rate was reduced to 0.1 C after 100

cycles, with a slight voltage fade ( < 0.05 V; Fig. 2g), again indicating the structural robustness of the composite structure in NM9505-0.95Li. In contrast, fast capacity decay (50 mAh/g) and voltage fade ( > 0.1 V) were observed in NM9505-1.05Li over the 100 cycles.

In addition to the NM9505 composition, we attempted to increase Mn substitution up to 10%, with the main results provided in Fig. S6. NM9010 exhibited up to 83.8% capacity retention in NM9010 at 0.5 C for 100 cycles, surpassing the 77.2% retention in NM9505 under the same cycling conditions. However, the initial capacity in NM9010 was much lower, only 198 mAh/g at 0.1 C and 160 mAh/g at 0.5 C. Similar observations are reported in the literature [18], indicating that low Mn is preferred in Ni/Mn-based Co-free CAMs in achieving both high capacity and high cycling stability. Therefore, our efforts were focused on the NM9505 composition and its structural tuning through Li stoichiometry control.

## Structural and morphological tuning of NM9505 via Li stoichiometry

The impact of Li stoichiometry on the structure, particle morphology, and electrochemical performance was systematically studied using a series of samples prepared in Li-deficient (0.90 and 0.95Li), near-stoichiometry (1.0 and 1.025Li), and Li-excess (1.05 and 1.10Li) conditions. The chemical compositions were analyzed using inductive coupled plasma-atomic emission spectrometry (ICP-AES), showing an overall agreement with the targeted stoichiometry (Table S1; Supplementary Note 4). With different Li stoichiometries, the obtained NM9505 powders exhibited markedly different primary particle morphologies, although they shared a similar morphology for the secondary particles. This distinction is clearly shown by SEM images (Figs. 3a and S7). Primary particles were small in the Li-deficient CAMs, exhibiting an elongated rod shape that resembled the precursors (Fig. S8). In contrast, the cuboid shape with increased particle size was found at near-stoichiometry and Li-excess conditions. The quantitative comparison is shown by the statistics of particle-size distribution in Fig. 3b, with measurements conducted on the smallest dimension of more than 100 primary particles (Fig. S7). The difference is substantial as Li-deficient NM9505 particles were small, below 100 nm (with a narrow distribution, marked by the shaded region), compared to those near-stoichiometry and Li-excess ones (up to a few hundred nm with a wide distribution). This observation suggests the critical role of Li stoichiometry in tuning the shape and size of primary particles (Supplementary Note 5).

The impact of Li stoichiometry on the structural ordering in the NM9505 series was investigated through synchrotron XRD analysis by examining the shape change of diffraction peaks as Li content increased from 0.90Li to 1.10Li (Fig. 3c). For example, all the diffraction peaks overall became sharper while the separation between the (110) and (108) peaks associated with the layered phase was enlarged as Li/TM ratio increased, up to 1.00, suggesting the formation of a well-ordered layered structure[19]. Note here that the (104) peak is a superposition of $(104)_L$ and $(200)_{RS}$ in the 2-phase composite, shifting to higher angles as Li increased, as shown in Fig. S2c. The (104) peak shift was less significant when Li further increased beyond 1.025, suggesting the least $Ni^{2+}$ formation when there was sufficient Li. However, the extra Li caused the formation of $Li_2CO_3$ residual on the particle surface (Fig. S9, Supplementary Note 6).

Quantitative analysis on the structural evolution of NM9505 as a function of Li stoichiometry was made through refinement, with main results provided in Figs. 3d, S10, S11 and Table S2 in the SI. The structure of the NM9505 with 1.05 and 1.10 Li can be well-refined using a layered phase. However, for the samples with Li content less than 1.025, the two phases coexisted, with the RS phase fraction decreasing dramatically as Li content increased, from 23% to nearly 3% when Li content reached 1.025 (Fig. 3d). The results highlight the strong dependence of RS content on Li stoichiometry. Due to the

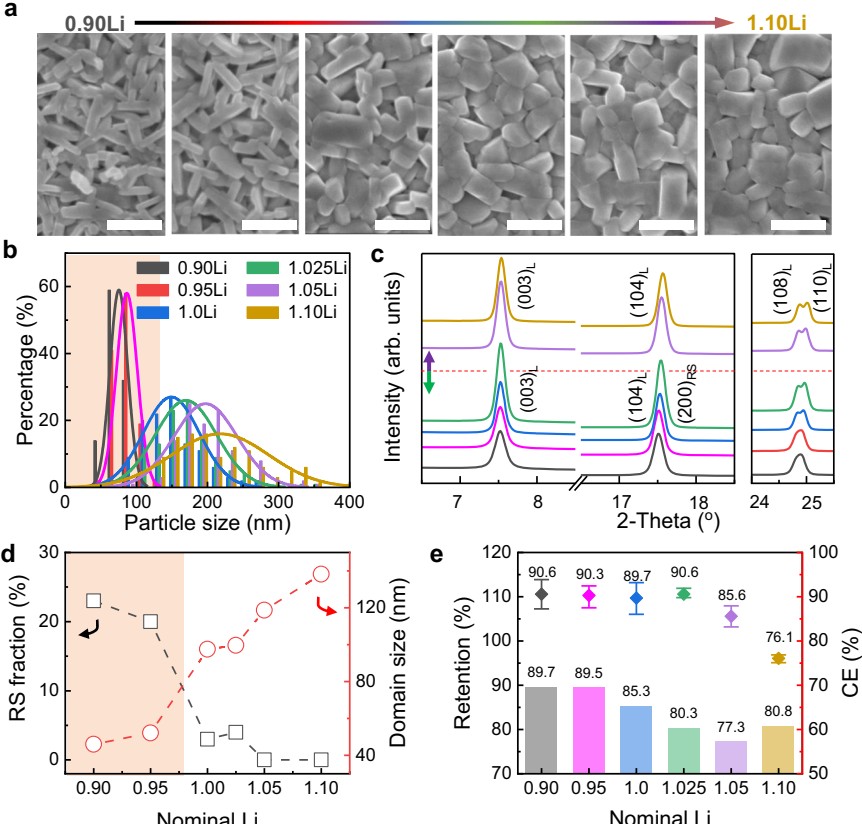

**Fig. 3 | Strong dependence of structure and morphology on Li-stoichiometry in NM9505. a** SEM images of the NM9505 calcined from the precursors with Li stoichiometry ranging from 0.90 to 1.10 (from left to right), demonstrating the strong dependence of the primary particle morphology on Li stoichiometry. Scale bars: 500 nm. **b** Particle size distribution of NM9505, showing the small size (below 100 nm) and narrow size distribution of the calcined NM9505 with Li stoichiometry below 1.00 (Li deficient, as shadowed), in contrast to the large size and wide size distribution of the samples with near stoichiometry and Li excess. **c** Synchrotron XRD patterns of NM9505, illustrating the evolution of the structural ordering with increasing Li stoichiometry ($\lambda = 0.6199$ Å). **d** Calculated phase fraction and domain size of the layered phase. **e** First-cycle CE (average of six cells, the error bar represents standard deviation) and capacity retention of NM9505 with different Li stoichiometry after 100 cycles at 0.5 C, showing higher values in the Li-deficient ones (0.9Li and 0.95Li), compared to those with Li excess (1.05Li and 1.10Li).

presence of $Ni^{2+}$ induced by the 5% Mn, the Li/TM ratio in the structure is less than one and consequently, Li/Ni mixing was present even when the Li source was in excess. The XANES and two-dimensional TXM mapping at the Ni K-edge further confirmed the little change of the Ni valance state as Li content went beyond 1.00 (Figs. S12 and S13 and Notes 7 and 8).

An important cell parameter in the layered phase is the Li slab thickness, serving as an indicator of the structural ordering of the layered phase. Interestingly, Li-deficient samples displayed a larger Li slab (Table S2) because there is less Li/Ni mixing in those layered domains in the composite-structured NM9505. In addition, the domain size of the layered phase increased with the rise of Li content, as displayed in Fig. 3d, highlighting the critical role of lithiation in driving crystal growth.

The electrochemical performance of the NM9505 CAMs with varying Li content was measured for comparison (Figs. 3e and S14). In the range of Li-deficient and near-stoichiometry, the first-cycle discharge capacity increased with rising Li content, and they all possessed a consistently high CE of nearly 90%. This may be explained by the fast Li intercalation kinetics in small primary particles[44]. In contrast, the first-cycle discharge capacity and CE experienced a significant decrease in Li-excess ones due to the resistive surface layer formed from residual Li[33]. Overall, NM9505 demonstrated high capacity retention under Li deficiency, and as excess Li was added, its cycling performance showed deterioration.

## Impact of Li-stoichiometry on the crystallization process during calcination

Although the morphology and size of primary particles differed significantly among the final NM9505 with different Li stoichiometries (Fig. 3a), they exhibited quite similar morphology and size after calcination at 600 °C for 12 hours (Fig. S15). Moreover, they retained the rod shape, resembling the primary particles in the precursors (Fig. S8). To delve into the impact of Li-stoichiometry on the phase propagation and crystallization, we investigated the calcination processes of NM9505-0.95Li and 1.05Li using time-resolved in situ XRD. For these in situ experiments, the mixtures of NM9505 hydroxide precursors and the nominal amount of $LiOH\cdot H_2O$, corresponding to 0.95Li and 1.05Li, were subjected to a two-step heating process: at 600 °C for 2 hours and then 720 °C for 3 hours (Figs. 4a–d and Fig. S16).

NM9505-0.95Li and NM9505-1.05Li underwent overall similar physico-chemical processes, involving (a) removal of water from the TM hydroxides, (b) oxidation and lithiation of the hydroxide precursors by reaction with O and Li salt, and (c) mass transport (including all species, such as TM, Li, and O) resulting in sintering of the particles. In both cases, an intermediate Li-containing RS formed and then gradually transformed into the layered phase. Upon reaching 600 °C, a high RS content was maintained and the layered domains remained small, suggesting sluggish crystallization kinetics at low temperatures. This observation is consistent with the SEM findings (Fig. S15): all samples of vastly different Li contents exhibited a similar morphology,

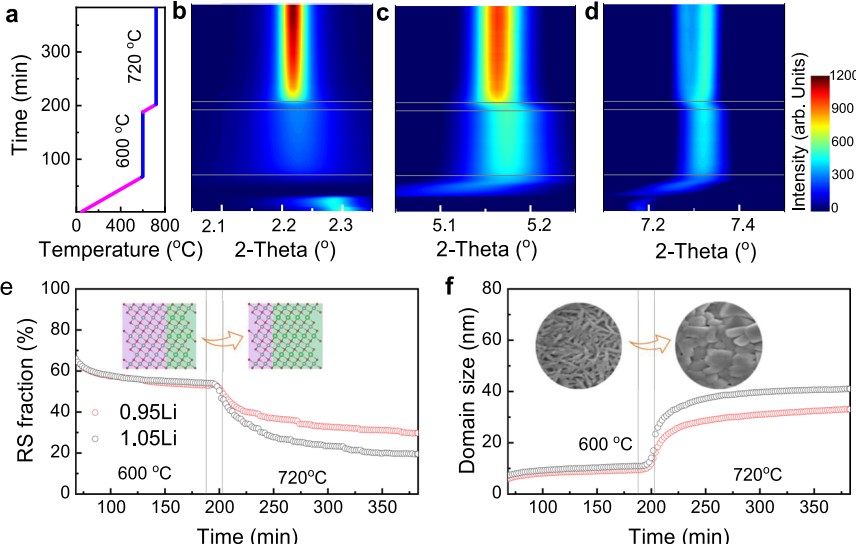

**Fig. 4 | Impact of Li stoichiometry on phase progression and crystallization during calcination of NM9505.** **a** Heating profile of the in situ XRD experiment. **b–d** Contour plots of the time-resolved in situ XRD patterns during calcination of NM9505-0.95Li (see Fig. S16 for the data from NM9505-1.05Li). **e** Calculated RS phase fraction as a function of time during calcination of NM9505-0.95Li (red) and NM9505-1.05Li (black), respectively. Inset: structure evolution from RS (purple shading) dominated to layered phase (green shading) dominated during calcination (Red: Oxygen, Green: Li, Gray: TM). **f** Crystal growth of the layered phase as a function of time for NM9505-0.95Li (red) and NM9505-1.05Li (black), respectively. Inset: SEM images of NM9505-1.05Li at 600 °C and 720 °C.

maintaining a small-sized rod shape even after 12 hours of sintering at 600 °C. As the temperature increased further, the RS proportion dropped quickly, turning into the layered phase due to thermally driven Ni oxidation and Li incorporation (Fig. 4e, inset). Consequently, the layered domain grew abruptly (Fig. 4f, inset). The kinetics of structural ordering and crystal growth strongly depended on the Li stoichiometry at elevated temperatures, as shown in Figs. 4e and 4f. The transformation from RS to the layered phase went much faster in NM9505-1.05Li compared to NM9505-0.95Li, resulting in lower RS fraction and larger-sized layered domains, which is consistent with the ex situ results (Fig. 3d). The role of Li stoichiometry in facilitating structural ordering and crystal growth will be discussed below, in correlation with multiscale modeling.

## Discussion

In correlation with multiscale experimental observations, computational modeling was performed regarding the calcination process to better understand the impact of lithium stoichiometry on phase propagation and crystallization, and consequently, the size and size distribution of the primary particles in the final CAMs. Details of the simulations, techniques, and further discussions are provided in Fig. S17, Notes 9 and 10 (SI). The primary particle microstructure of the NM9505 hydroxide precursors, consisting of spherical particles, was computationally generated in two dimensions with an average particle diameter of around 35.0 nm (Fig. 5a). The primary particle microstructures of the calcined NM9505 at different temperature and Li/TM ratios were computationally simulated, with some examples given in Figs. 5b–d. Figure 5e summarizes the final computational results and compares them with the experimental observations. It is evident that the rate of increase in particle size is low for a Li/TM ratio less than unity (Li/TM < 1), while a significantly larger particle growth rate is observed at more than stoichiometric amount of Li (Li/TM > 1).

We propose that primary particle growth is mediated by mass transfer, through two mechanisms, namely, lithiation-induced crystallization and liquid phase sintering. As demarcated in Fig. 5e with a vertical line at Li/TM = 1, there exist two distinct domains, where lithiation-induced crystallization is dominant, and where liquid phase sintering plays a major role. This provides a reasonable explanation for

the experimental observations, namely the abrupt size change at Li/TM = 1. The particle growth mechanisms dominated by lithiation-induced crystallization and liquid phase sintering are simplified in Figs. 5f and 5g. When the Li source is just enough or below the stoichiometry (Li/TM < 1.0 or ~ 1.0), most Li is incorporated into the particles, participating in phase transformation and crystal growth during calcination, resulting in smaller-sized composites consisting of RS and layered phases (Fig. 5f). When extra Li is added (Li/TM > 1.0), not all the LiOH reacts with the cathode precursors during calcination, and extra lithium salt exists in a molten state surrounding the primary particles (Fig. 5g). A liquid phase sintering mechanism gets activated due to the faster mass diffusion through this liquid phase. Therefore, small primary particles tend to merge into larger ones, similar to the synthesis of single-crystal NMC cathode materials via the molten salt method[45], which is sometimes characterized as the Ostwald ripening process. Consequently, the overall primary particle size ends up being large under Li-excess conditions.

As reported in the literature, the primary particle size is impacted by various factors during calcination, including chemical composition, temperature, and oxygen pressure[35,46–48]. We demonstrate here the strong dependence of primary particle size on Li content in the precursors (and so the Li stoichiometry in the synthesized CAMs), which is corroborated by multiscale modeling, providing insights into the role of Li stoichiometry in governing phase progression and crystallization.

The Li-deficient NM9505 possesses a smaller primary particle, consisting of intergrown layer and RS phases, which minimizes the stress generated by volume change during the Li insertion and extraction, which in turn mitigates microcracking during long-term cycling[44,49,50]. Furthermore, the refined primary particle size could improve the Li transfer kinetic and reduce the first-cycle loss. Compared to the recent reports on reducing particle size through coating/doping, using tungsten and molybdenum[15,51,52], this method of using the Li stoichiometry as a tuning handle provides a more cost-efficient and effective strategy. Along with the primary particle morphology, the unique intergrown composite structure with RS and layered phase in the same *ccp* oxygen framework plays an important role in structural stabilization, by suppressing the anisotropic strain of the layered structure[36,43].

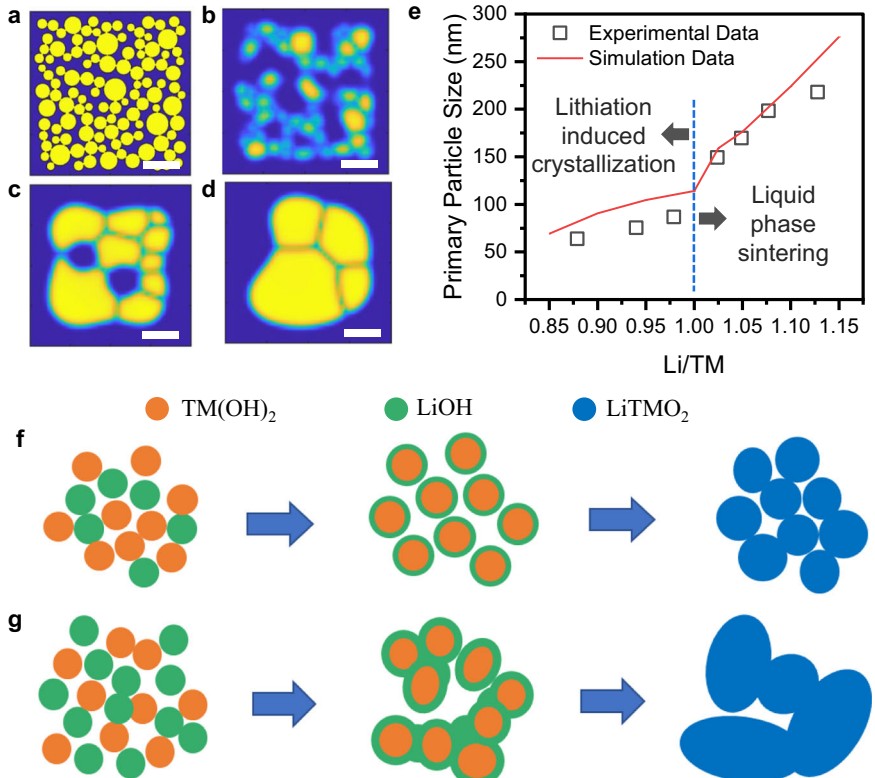

**Fig. 5 | Mechanistic understanding of the roles of Li stoichiometry in tuning crystal growth and inter-particle fusion during the calcination process. a** Initial microstructure of the primary particles with an average size of ~35 nm (measured by SEM; Fig. S8). Scale bars: 100 nm. **b** Computationally simulated primary particle microstructure after calcination at 600 °C for 12 hours (Li/TM = 1.0), with an average particle size similar to the initial value (consistent with SEM observation as given in Fig. S15 (**a**)). **c**, **d** Computationally simulated primary particle microstructure after calcination at 720 °C for 12 hours with a Li/TM ratio of 0.95 and 1.05, respectively. **e** Comparison between the experimentally observed (black squares) and computationally predicted primary particle size (red line) as a function of the Li/TM ratio obtained after calcination at 720 °C for 12 hours. The two distinct regions, where lithiation-induced crystallization is dominant, and where the liquid phase sintering becomes predominant, are clearly demarcated. **f**, **g** Schematic representation of the sintering induced particle growth mechanism observed during the calcination of Ni-rich cathode primary particles with different amounts of lithium salt. TM(OH)₂ precursors, LiOH salt particles, and lithiated LiTMO₂ particles are denoted by orange, green and blue, respectively. Scheme (**f**) indicates that, under lithium poor condition (Li/TM < 1.0) or stoichiometric amount (Li/TM ≈ 1.0), all the lithium is consumed in the reaction with the cathode precursors, and with no excess lithium salt, no fusion among individual grains occurs during calcination. On the contrary, scheme (**g**) indicates that with the presence of excess lithium salt (Li/TM > 1.0), a significant amount of the non-reacted molten lithium salt exists adjacent to the cathode primary particles. This excess lithium salt in liquid phase acts as a sintering aid and leads to a substantial amount of sintering-induced particle growth through inter-particle fusion.

During charging/discharging processes, the H2 to H3 phase transition is largely suppressed and no abrupt layer shrinkage is observed, even in a deeply charged state[25]. As shown in Fig. S18, the phase fraction of RS in NM9505-0.95Li is calculated to be 17.5% after 100 cycles, slightly smaller than the percentage of the RS in the pristine sample (19.6%). This indicates that the layered/RS composite structure can be well retained. There have been different interpretations regarding the degradation over cycling. Yabuuchi et al. showed the Ni migration to the tetrahedral sites in the over-delithiated NiO₂[53]. Due to the different material systems and the cycling conditions, the degradation mechanism can be very different, but eventually, the structural degradation leads to the formation of the electrochemically inactive RS. Note that the RS in the Li-deficient NM9505 is uniformly distributed across the whole particle, intergrown with the layered phase in the composite (Fig. 2a–d). This sets it apart from the RS as structural defects caused by surface reconstruction during electrochemical cycling[7,54].

In summary, we demonstrated rational control over Li-stoichiometry to enable NM9505 as a Co-free CAM for practical use in next-generation LIBs. Remarkably high performance was obtained in the 5% Li-deficient NM9505 with a layered-RS composite structure, outperforming Li-excess counterparts by delivering high capacity retention (90% after 100 deep cycles), high first-cycle CE (> 90%), and

negligible voltage decay. The crystallization thermodynamics and kinetics during calcination of NM9505 were investigated through multiscale-correlated characterization and modeling, revealing the crucial role of Li stoichiometry in governing the kinetics of structural ordering and crystal growth. Specifically, liquid-phase sintering occurred in the presence of excess Li, leading to large primary particles in CAMs. In contrast, small primary particles consisting of intergrown RS and layered phases were formed in Li-deficient CAMs; due to the small anisotropic lattice expansion and contraction during cycling, they exhibited high structural stability. These findings demonstrate an efficient route to structural and morphological tuning via Li stoichiometry for stabilizing Co-free cathodes, with desired cost-efficiency achieved by eliminating the extra coating/doping steps[37,38,55]. By offering a promising and cost-effective alternative to Co-reliant cathodes, our study addresses the critical need for the development of sustainable and commercially viable LIBs.

## Methods

### Synthesis of NM9505 with different Li stoichiometries

Ni₀.₉₅Mn₀.₅(OH)₂ precursor powders were synthesized continuously using a Taylor Vortex Reactor via hydroxide coprecipitation[56–58]. In detail, the NiSO₄·6H₂O and MnSO4·H2O were used as starting

materials and dissolved in nitrogen-purged DI water to form a 2 M solution with a Ni:Mn molar ratio of 95:5. A 4 M solution of NaOH was employed as co-precipitating agent, while the chelating agent was 4 M $NH_4OH$. The solution's pH and temperature in the reactor were maintained at 11.92 ($\pm 0.02$) and 52 °C ($\pm 0.2$ °C), respectively. The inner cylinder was rotated at 800 rpm, creating a Taylor vortex flow pattern where the reaction occurred. The resulting product was thoroughly washed with nitrogen-purged DI water and then dried in vacuum overnight at 120ºC. The obtained hydroxide precursor was then mixed with $LiOH \cdot H_2O$ (Sigma-Aldrich, 99.95% pure) at different molar ratios (Li: (Ni+Mn) = 0.90, 0.95, 1.00, 1.025, 1.05, and 1.10). The mixed powders were calcinated in a tube furnace (MTI) at 600 °C for 12 hours and then at 720 °C for 12 hours under a continuous $O_2$ flow, with 1 L/min $O_2$ flow rate.

## Sample characterization

The chemical composition of the NM9505 was determined using inductively coupled plasma mass spectrometer (ICP-MS, Agilent 7700 Series). The particle morphologies were observed by scanning electron microscope (JOEL-7600F). STEM images with HAADF detector were obtained using a JEOL ARM 200CF microscope equipped with two aberration-correctors and a cold-field-emission electron source. High-resolution X-ray diffraction was performed at beamlines 7-BM(QAS) and 28-ID-2(XPD); X-ray absorption spectroscopy was performed at beamline 7-BM (QAS); and transmission X-ray microscopy was performed at beamline 18-ID (FXI) at the National Synchrotron Light Scours (NSLS-II) at Brookhaven National Laboratory. For in situ XRD, the $Ni_{0.95}Mn_{0.05}(OH)_2$ precursors were mixed with $LiOH \cdot H_2O$ at molar ratios of 0.95 and 1.05, respectively. The mixture was pressed into pellets, loaded into a Linkam furnace (TS1500), and then mounted to the beamline (28-ID-2). XRD patterns were continuously recorded as the samples were heated from room temperature to the desired temperature (600 °C and 720 °C) at a heating rate of 10°C/min. The heating profile is shown in Fig. 4a.

## Electrode fabrication and electrochemical testing

The obtained NM9505 cathode powders were mixed with Super P carbon and polyvinylidene fluoride at a ratio of 8:1:1 and dissolved in N-methyl-2-pyrrolidone. To obtain a uniform slurry, the mixture was further mixed using a planetary centrifuge mixer (Thinky). The slurry was then cast on carbon-coated Al foil and dried in a vacuum oven at 100 °C overnight. The active materials loading of the electrodes was 7–8 mg/cm². The electrode was punched into 10-mm-diameter disks for coin cell fabrication. Li metal was used as the counter electrode and $LiPF_6$ in a mixture of EC/DEC was used as the electrolyte. All the cell fabrication steps were conducted in an Ar-filled glovebox with $O_2$ and $H_2O$ levels less than 0.1 ppm. The battery cycling was tested using the Landt battery test system at room temperature ( ~ 22 °C). The coin cell was first cycled between 2.7 and 4.4 V at 0.1 C (1 C = 200 mA/g) for three cycles and then 0.5 C charge/discharge rate for the rest cycles.

## Computational analysis

Details of the calcination simulations conducted at the mesoscale level is provided within the supporting information section (see Supplementary Notes 9 and 10 in SI).

# Data availability

Source data has been provided. Additional data related to the study are available from the corresponding author upon reasonable request. Source data are provided with this paper.

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

## Acknowledgements

Research at Brookhaven National Laboratory is supported by the U.S. Department of Energy Office of Energy Efficiency and Renewable Energy Vehicle Technologies Office under Contract No. DE-SC0012704. This research used the 7-BM(QAS), 18-ID (FXI) and 28-ID-2(XPD) beamline of the National Synchrotron Light Source II, a DOE Office of Science User Facility operated for the DOE Office of Science by Brookhaven National Laboratory under Contract No. DE-SC0012704. Efforts on STEM characterization were supported by the U.S. DOE, Office of Basic Energy Science, Division of Materials Science and Engineering (contract no. DE-SC0012704). SEM measurements carried out at the Center for Functional Nanomaterials, Brookhaven National Laboratory, were supported by the DOE, Office of Basic Energy Sciences, under Contract No. DE-SC0012704. Research at Argonne National Laboratory is supported by the U.S. Department of Energy Office of Energy Efficiency and

Renewable Energy Vehicle Technologies Office under Contract No. DE-AC02-06CH11357. The baseline precursor material reported in this paper was synthesized at the Materials Engineering Research Facility (MERF), Argonne National Laboratory. The MERF was supported by the DOE, Office of Energy Efficiency and Renewable Energy, and the Vehicle Technologies Office.

## Author contributions

K.C., J.B. and F.W. conceived the ideas and designed the experiments. K.C. synthesized the materials, carried out electrochemical measurements and characterizations, and analyzed the data. P.B. and V.S. performed the modeling work. O. K and K.Z.P synthesized precursors and ran ICP. L.W. and Y.Z. performed TEM and data analysis. M.G. set up the TXM experiments and helped with TXM data analysis. L.M. and S.N.E. setup the XRD and XANES experiments and helped with XRD and XANES data analysis. K.C., F.W. and J.B. performed the in situ and ex situ XRD. J.B. and H. Z. analyzed the XRD data. K.C., F.W, P.B. and J.B. wrote the manuscript with input from all co-authors. All authors commented and reviewed the manuscript. J.B. and F.W. supervised the work.

## Competing interests

The authors declare no competing interests.
