## [Peer Review File · Nature Communications]

Cobalt-Free Composite-Structured Cathodes with Lithium-Stoichiometry Control for Sustainable Lithium-ion BatteriesREVIEWER COMMENTS

Reviewer #1 (Remarks to the Author):

This paper provides the impact of particle size and lithium stoichiometry on electrode performance of layered materials, $\text{Li}_x\text{Ni}_{0.95}\text{Mn}_{0.05}\text{O}_2$. This paper also shows the interesting data, but the analysis provided is not clear without rational explanations or a lack of experimental data. Specific comments are described in the following section:

(1) Co-free sample is not innovative enough because Y. K. Sun and J. R. Dahn's group have already reported Co-free layered system with high performance.

(2) The most important part is that the structural analysis is not clear. From the data shown in this article, I cannot accept the logic of composite structure of layered phase and rocksalt phase, which was derived only from XRD data. Authors must provide the direct evidence of the composite structure from other spectroscopic data (cross-sectional imaging by STEM/EDX etc.). From XANES data, the rocksalt domain must consist of similar structure with NiO. Because if the rocksalt phase has the chemical composition of $\text{Li}_{0.5}\text{Ni}_{0.5}\text{O}$, Ni must be trivalent state regardless of the difference in structures. However, the lattice parameter of the rocksalt phase is clearly different from NiO. Moreover, the non-uniform distribution of chemical compositions as the composite structure are not observed in TXM mapping.

Instead of the composite structure proposed in this paper, I think that a Li deficient phase, is more suitable model, and $(\text{Li}_{0.9})_3\text{a}(\text{Ni}_{0.95}\text{Mn}_{0.05})_3\text{bO}_2$ can be reformulated to $(\text{Li}_{0.947}\text{Ni}_{0.053})_3\text{a}(\text{Ni}_{0.947}\text{Mn}_{0.053})_3\text{bO}_2$ as a single layered phase with partial cation disorder. Ni average oxidation state is 2.84 for this phase. Note that the presence of Li at 3b sites is ignored for simplicity in this model, and the presence of Li in 3b sites can be analyzed by neutron diffraction study. From the XRD data, and above-mentioned points, I think that the partial cation disorder is more likely scenario. Please carefully analyze the experimental results.

(3) Figure 4, the RS fraction also must have the problem as analysis. The data of Figure S15 is also analyzed as the layered structure with partial Li/Ni disorder.

(4) Figure S16, this analysis also may have the problem. Please check the publication of Yabuuchi et al. (J. Mater. Chem. A, 2021, 9, 15963–15967). Figure 3a in the Yabuuchi's publication also have similar data with this publication, but they analyzed as the Ni migration to tetrahedral sites in Li layer.

(5) Authors claims that "small anisotropic lattice expansion and contraction during cycling, they exhibit high structural stability". However, the direct evidence was not provided.

(6) Figure 5 f, g is not clearly described. I think that if the composite model was used similar crystallization process is expected for $\text{LiNi}_{0.95}\text{Mn}_{0.05}\text{O}_2$ and $\text{Li}_{0.90}\text{Ni}_{0.95}\text{Mn}_{0.05}\text{O}_2$.

Reviewer #2 (Remarks to the Author):

In this manuscript, the authors investigated the impact of $\text{LiOH}/\text{M}(\text{OH})_2$ molar ratio on the primary particle growth and the electrochemical performance of high-Ni, Co-free cathode materials. I think this is an interesting work done by a group of well-respected battery research scientists. However, some major changes and clarification are needed before this manuscript may be considered for publication at Nat. Commun.

(1) More electrochemical tests should be carried out to demonstrate the advantage of the Li-deficient Ni-rich cathode. What about the rate-performance, Li diffusivity, full cell

performance? The authors only showed half-cell performance from one cell in each case. This is not convincing enough if this manuscript is going to be published at Nat. Commun. If we only compare the half-cell performance, I think the 1.022Li-sample looks very good. Then, what is the point of using Li-deficient Ni-rich cathode?

(2) More structural characterizations may be needed. The authors claimed a minor Li-containing rock-salt phase ($\text{Li}_y\text{TM}_{1-y}\text{O}$) of about 19.6 mol%. This is a significant amount. Can it be observed via STEM-HAADF?

(3) What is the exact value of y in $\text{Li}_y\text{TM}_{1-y}\text{O}$? When the authors carried out the Rietveld refinements, did they consider other possible "second phases?" Can it be a Li-deficient layered phase? If it is really a Li-containing, NiO-like rocksalt, wouldn't it negatively impact the Li transport and the rate performance? This is why I suggest GITT measurements in my first comment.

(4) The sintering time appears to be quite long, 600 C for 12h and then 720 for 12h. What is the point of having the 600 C step? The oxygen flow rate should be reported.

(5) The particle size distribution is obtained from the SEM images of the secondary particle surface. What about the interior of these secondary particles? Can the authors carry out some ion-milling or FIB experiments to check the inside?

(6) In the abstract, the authors write "the anisotropic lattice expansion and contraction is suppressed in Li-deficient NM9505 upon cycling". I did not see any in situ electrochemistry-XRD data to support this claim.

(7) Fine-tuning the composition and microstructure of the layered cathode materials seems an interesting new trend. How does the Li-deficient Ni-rich, Co-free cathode compare with the slightly Li-rich nickel oxide cathode and the Li-rich, Ni-rich $\text{Li}_{1.09}\text{Ni}_{0.85}\text{Mo}_{0.08}\text{O}_2$ cathode?

Reviewer #3 (Remarks to the Author):

The present manuscript with title "stabilising cobalt-free cathodes with lithium-stoichiometry control for sustainable lithium-ion batteries" by Chen et al. explores the effects of Li stoichiometry in the structure and electrochemical properties of Co-free $\text{Li}_x\text{Ni}_{0.95}\text{Mn}_{0.05}\text{O}_2$ (NM9505) using synchrotron X-ray diffraction and multiscale modelling as main characterisation tools. Using as a proof of concept materials Li-deficient (0.95) NM9505 they show that Li deficiency in the structure is key to hindering crystallization and interparticle fusion during calcination, leading to the formation of composites containing intergrown layered and rocksalt phases which are responsible for good performance of this material with respect to the stoichiometric material in terms of ICE, capacity retention and negligible voltage fade. The work is relevant to the battery research and industry community and therefore, we would like to recommend the manuscript to be accepted after some major corrections.

- The composition of the RS phase observed in the Li-deficient samples is not entirely clear. What RS phase was used in the Rietveld refinement? More detail should be given about what the stoichiometry of this RS might be, even if this is tentative.
- 3D TXM mapping for the Li stoichiometric phase should be provided for reference so one

can understand how these data compare to the 3D TXM mapping provided for the Li-deficient phase. It should be clarified in the manuscript reason for having both Ni³⁺ and Ni²⁺ ions present in the Li-deficient material.

- In line 121 the authors mention a slight overpotential in the Li-deficient NM9505 sample. It should be indicated in which potential window this is observed. This is explained with the presence of RS but no more explanations are provided. The authors should clarify this statement.

- In line 125 (Figure S4), the authors should highlight the different H1, M, H2 and H3 phase transitions in the Figure.

- Figure 2c- different colour choices should be made to observe differences in oxidation state for Ni.

- In line 154, the authors highlight that the initial capacity of NM9010 is 160 mAh g⁻¹. However, Figure S5 shows that this capacity is close to 200 mAh g⁻¹. There is a large decay in capacity after 3-4 cycles from this initial capacity to 160 mAh g⁻¹ which the authors have not mentioned. How many repetitions of the electrochemistry in this material have been made? Is this common in all repetitions? If so, an explanation should be provided as to why this is happening.

- Microscopy images of the different samples after long cycling should be provided to confirm that there is less microcracking on the Li-deficient NM9505.

- In Figure S9. Li₂CO₃ peaks should be indicated in the Figure.

- In Figure S11, references for the different oxidation states of Ni and Mn should be provided to allow a better comparison between these and the data provided in the manuscript.

- Some typos were found in the manuscript: e.g. Line 72 "Materials", Line 144 "valance", Line 194 "stoichiometry..Even"

One-to-one Response

Reviewer #1

This paper provides the impact of particle size and lithium stoichiometry on electrode performance of layered materials, $\text{Li}_x\text{Ni}_{0.95}\text{Mn}_{0.05}\text{O}_2$. This paper also shows the interesting data, but the analysis provided is not clear without rational explanations or a lack of experimental data. Specific comments are described in the following section:

We appreciate the reviewer for the comments with great insights and detailed explanation. By following the reviewer's comments, we have re-examined our data analysis and interpretation. Significant revisions were made to the manuscript, along with new experimental data added to support our main conclusions. We believe we have addressed the major concerns from the reviewer, as detailed below.

(1) Co-free sample is not innovative enough because Y. K. Sun and J. R. Dahn's group have already reported Co-free layered system with high performance.

Response: We agreed with the reviewer that significant work has been reported in Co-free layered system over the decades, as in Y.K Sun and J.R. Jahn's groups (being cited in this manuscript). It should be noted that, so far, the reports have been focusing on the layer-structured systems, with their performance improved by reducing the cationic disordering (*i.e.*, Li/Ni mixing) through synthesis/processing coupled with compositional tuning and coating(doping).

Despite the efforts over the decades, Co-free *layer-structured* cathodes have not been commercialized yet, because of the fundamental challenge associated with Co elimination, namely, the unavoidable disordering and cycling instability. Instead of making the traditional layer-structured Co-free cathode systems, we synthesized the Co-free cathodes ($\text{Li}_x\text{Ni}_{0.95}\text{Mn}_{0.05}\text{O}_2$), with a composite structure, through the Li stoichiometry control (*as to be explained below in responses to the reviewer's following comments*).

Simply put, the key innovation of our work is to stabilize the Co-free cathodes through Li-stoichiometry control. Indeed, we demonstrated the Li-deficient composite systems can outperform those Li-stoichiometric layered counterparts due to their unique structure, namely the composite structure, consisting of the intergrown rocksalt and layered phases.

Revisions: to further clarify the innovations of this work, we have made changes to the Abstract by stressing the challenges associated with eliminating Co from the layered NMC and re-stating our approach:

... Despite numerous attempts to address this challenge, eliminating Co from layer-structured cathodes remains elusive, as doing so detrimentally affects their layering and cycling stability. Herein, we report rational control over Li-stoichiometry in synthesis of $\text{LiNi}_{0.95}\text{Mn}_{0.05}\text{O}_2$ (NM9505), resulting in Li-deficient composites consisting of intergrown layered and rocksalt phases that outperform traditional layered counterparts.

(2) The most important part is that the structural analysis is not clear. From the data shown in this article, I cannot accept the logic of composite structure of layered phase and rocksalt phase, which was derived only from XRD data. Authors must provide the direct evidence of the composite structure from other spectroscopic data (cross-sectional imaging by STEM/EDX etc.). From XANES data, the rocksalt domain must consists of similar structure with NiO. Because if the rocksalt phase has the chemical composition of $\text{Li}_{0.5}\text{Ni}_{0.5}\text{O}$, Ni must be trivalent state regardless of the difference in structures. However, the lattice parameter of the rocksalt phase is clearly different from NiO. Moreover, the non-uniform distribution of chemical compositions as the composite structure are not observed in TXM mapping.

Instead of the composite structure proposed in this paper, I think that a Li deficient phase, is more suitable model, and $(\text{Li}_{0.9})_3\text{a}(\text{Ni}_{0.95}\text{Mn}_{0.05})_3\text{bO}_2$ can be reformulated to $(\text{Li}_{0.947}\text{Ni}_{0.053})_3\text{a}(\text{Ni}_{0.947}\text{Mn}_{0.053})_3\text{bO}_2$ as a single layered phase with partial cation disorder. Ni average oxidation state is 2.84 for this phase. Note that the presence of Li at 3b sites is ignored for simplicity in this model, and the presence of Li in 3b sites can be analyzed by neutron diffraction study. From the XRD data, and above-mentioned points, I think that the partial cation disorder is more likely scenario. Please carefully analyze the experimental results.

Response: We thank the reviewer for explaining his/her concerns in detail. Indeed, structural analysis by XRD for such a complex system is challenging particularly because of the large overlap of the diffraction peaks associated with lithiated rock-salt (RS) phases with those associated with layered phase. We'd like to address the reviewer's concerns with the newly added experimental results and elaborated details on data analysis, as explained below.

To discriminate the subtle structural difference between layered and RS, we took the high-quality data, with the desired high-intensity and high-resolution by using synchrotron source. We tried the two different structure models, single layered phase and two-phase model (with layered and RS composites), as shown in **Figure S2**. Clearly, the fitting to the experimental data using the 2-phase model is much improved compared to that using the single-phase (layered) model. Also importantly, those peaks at 15.2° (Figure S2c), 17.5° (Figure S2d), and 24.8° (Figure S2e) CANNOT be matched with the pure layered phase ALONE.

To better show the impact of the Li deficiency on the shape of diffraction peaks, we have added the XRD obtained from the NM9505 samples with 0.85 Li and 0.90Li, shown in **Figure R1**. As shown by the zoom-in view, those peaks at around 15.2° , 17.5° and 24.8° from NM9505 with 0.90 Li and 0.85 Li shows the similar behaviors as the data taken from NM9505 with 0.95 Li, but exhibiting even more pronounced difference (i.e., stronger sublet peaks associated with rocksalt phase). In contrast, the peaks from NM9505 with 1.05 Li are sharper, no sublet peaks associated with the RS phase. In addition, the presence of the RS phase caused the subtle, but noticeable shape change of the peaks associated with the layered phase, for example, making the peak at ~ 15.2 arising from (006)(102) of the layered phase broadened and more symmetric as Li content decreases because of the increased RS components. Similar broadening effect with the decrease of the Li content can be seen from two other peaks at around 17.5° and 24.8° . In contrast, in the

diffraction pattern from NM9505 with 1.05 Li, the peak at ~ 15.2 arising from (006)(102) of the layered phase is sharp and asymmetric due to the different intensity of the two peaks.

Figure R1 Synchrotron XRD of NM9505 with 0.85Li, 0.9Li, 0.95Li and 1.05Li.

We also realized the XANES data (Figure S11) may be misleading, WITHOUT the reference spectra. As shown in the Revised Figure S11 (Fig.S12 in the revision), the Ni valence is not fully oxidized into trivalent, (as the reviewer assumed: $\text{Li}_{0.5}\text{Ni}_{0.5}\text{O}$), but oxidized partially to 3 ($\text{Li}(\text{Ni}^{3+}_x\text{Ni}^{2+}_{(0.95-x)})\text{Mn}_{0.05}\text{O}_2$; as in the layered phase) and partially oxidized to a valence state between 2 and 3 ($\text{Li}_y(\text{Ni}^{3+}_y\text{Ni}^{2+}_{(1-2y)})\text{O}$; in the lithiated rock-salt).

Figure R2. X-ray absorption near-edge spectroscopy (XANES) of NM9505 obtained at different Li stoichiometry and the corresponding zoom-in view as marked in red square regions. NiO was used as references for Ni²⁺.

The reviewer is right that “*the lattice parameter of the rocksalt phase is clearly different from NiO*” and actually much smaller compared to that of NiO (being about 4.13, as reported in Steiger, Patrick, Ivo Alxneit, and Davide Ferri. "Nickel incorporation in perovskite-type metal oxides—implications on reducibility." *Acta Materialia* 164 (2019): 568-576.). As shown by the refinement result listed in Table S2; copied here, the Li-containing RS phase has a smaller lattice parameter as compared to the standard NiO, because it contains Ni³⁺. In addition, the measured Li occupancy in the RS phase is around 0.3, not 0.5.

Table S2. Refinement parameters of NM9505 with different Li contents.

Li/TM add	RS-mol	a(RS)	a-L	c-L	Li-slab	Ni-mix	L-domain size	c/a	Li Occupancy (RS)	Li/TMRwp
0.90	0.232	4.083	2.882	14.222	2.6675	0.008	45.98	4.934	0.358	0.916 2.35
0.95	0.196	4.082	2.881	14.219	2.6617	0.021	52.09	4.934	0.359	0.906 2.35
1.00	0.033	4.074	2.878	14.211	2.6371	0.035	97.53	4.936	0.24	0.917 2.06
1.025	0.037	4.070	2.876	14.205	2.6360	0.018	99.59	4.939	0.310	0.951 2.18
1.05	0	-	2.874	14.2015	2.6337	0.026	118.74	4.941	0	0.950 2.03
1.10	0	-	2.871	14.191	2.6336	0.022	138.18	4.943	0	0.956 1.91

Complementary to the synchrotron XRD analysis, the 3D TXM-XANES mapping provides information about local Ni²⁺ and 3+ distribution within individual particles. Although the resolution is relatively lower compared to the cross-section STEM-EDX, the TXM results are

statistically more meaningful and more importantly, valence state of Ni can be spatially resolved by TXM-XANES (not resolvable by EDX).

In response to the reviewer's comment (*the non-uniform distribution of chemical compositions as the composite structure are not observed in TXM mapping*), we added the zoom-in view for the local distribution of the two components (Ni²⁺ and Ni³⁺), in comparison to the TXM maps from the NM9505 with 1.05 Li, as shown in Figure R3. The non-uniform distribution of the chemical compositions is clearly shown in the TXM mapping of 0.95Li sample, in the form of “clusters” (red) across the secondary particles (associated with RS phase, with contrast from Ni²⁺) are clearly shown (**Figure R3** (a - c)). In contrast, no such contrast is shown in the TXM map from 1.05Li sample – overall uniform Ni²⁺ across the secondary particle (**Figure R3** (d-f)).

Figure R3, (a)(d)The center slide of the 3D TXM mapping of NM9505-0.95Li and 1.05Li. (b) and (e) are the zoom in views. Green and red color indicating Ni³⁺ and Ni²⁺ distribution, respectively. (c) (f) the distribution of Ni²⁺ of 0.95Li and 1.05Li.

By following the reviewer's comments, we have also taken STEM-HAADF images from the local region of the primary particles. Representative images from one primary particle are provided in **Figure R4**, showing the composite structure, consisting of intergrown layer and RS phases (as indexed in the FFT). See the description and explanation in the figure caption and notes below.

Figure R4. (a) STEM-HAADF image taken viewed along [100] direction from NM9505-0.95Li. The inset shows the low magnification image where the image in (a) is taken from the area indicated by the white arrow. (b-e) Magnified images (b,d) and their corresponding FFTs (c,e) from red and blue square in (a), showing rock-salt and mixture of rock-salt and layered structure, respectively. The [0-11] projection of the rock-salt and [100] projection of layered structure are embedded in the images. Orange, yellow, red and green spheres represent Li/TM, oxygen, Li and TM, respectively. The FFTs are indexed based on RS $(0-11)^*_{RS}$ and layered structure $(100)^*_L$, respectively. Subscripts RS and L represent RS and layered structure, respectively. (f-h) Magnified images from the area marked by rectangles in (b) and (d), and the area indicated by the green arrow in (a), respectively.

Note: The images in (b) and (f) show atomic columns with equal intensity, consistent with the rock-salt structure (see embedded [0-11] projection of rock-salt structure). The corresponding FFT in (c) is consistent with the $(0-11)^*$ diffraction pattern of rock-salt structure, further confirms the RS structure of this local area. The image in (h) shows layered structure where only TM atoms show contrast. The Li and oxygen atoms are invisible due to their low atomic number in Z-contrast STEM-HAADF image (Intensity approximately is proportional to $Z^{1.7}$, Z is atomic number). The images in (d) and (g) shows the layered arrangement with weak and strong contrast alternately arranged horizontally. We attribute this area to the mixture of RS and layered structure. The layered and RS structure overlapped along beam direction. The layers with strong contrast indicated by green arrows in (g) are overlap of TM layer (green spheres) of layered structure with Li/TM (orange) of RS, while those with weak contrast indicated by orange arrows are overlap of Li layer (red) of layered structure with Li/TM (orange) of RS. In FFT (e) from this area, the spots indicated by the black arrows are contributed by both layered structure and RS, thus show stronger intensity than those indicated by the red arrows which exclusively belong to the layered structure. This further confirms the mixture of layered structure and RS in this area. We took STEM-HAADF images from many places, showing the similar composite structure although the components of the layered and RS phases vary from region to region.

Revisions: to further improve clarify of the structural analysis, we have made the following changes:

- We added the synchrotron XRD data from NM9505 with 0.85, 0.90Li, 0.95Li and 1.05 Li in **Figure R1** into the supporting (as **Figure S2**) for to show the trend of changes in the phase components ...
- We re-drew the TXM data in **Figure 2(c)**, added with color bar and zoom-in views for clarification. We have also added the TXM mapping for the 1.05Li for comparison in **Figure S4**.
- We revised the **Figure S12**, added the reference XANES spectrum from the NiO standard.
- We added **Figure R4** into the supporting as Figure S4, and the representative STEM image as **Fig. 2c** in the main text.

(3) Figure 4, the RS fraction also must have the problem as analysis. The data of Figure S15 is also analyzed as the layered structure with partial Li/Ni disorder.

Response: As in the response to the previous comment, the two-phase model is more appropriate for fitting the final synthesized phase. Therefore, we have used the same model in the analysis of the in situ XRD data in **Figure 4**. As shown in Figure 4 (e), it is the RS phase that was dominant at low lithiation state (at 600 °C) and then the layered phase at high lithiated states (at 720 °C).

In order to show the trend of changes, we plot the zoom-in view of the diffraction peaks of the XRD data presented in in the **Figure R5**. It is clear that those peaks associated with the RS phase (labelled as “NiO”) are dominant at 600C and then decrease with holding.

Figure R5. Two-phase model for fitting the diffraction patterns taken at 660 °C (top) and the evolution of the phase fraction (bottom), showing the decrease of the proportion of RS (NiO) and the increase of the proportion of layered (Ni95) with time.

We understand the plot of the (003)/(104) in Figure S15 (f) is confusing as it has been commonly used to show the Li/Ni ordering in the layered structure. Actually, the purpose is to show the relative concentration/fraction of the layered phase, by using the fact that (003) is zero in the RS phase.

Revisions: In order to avoid confusions, we removed the Figure S15 (f) and (g) from the Figure S15 (Fig. S16 in the revised version).

(4) Figure S16, this analysis also may have the problem. Please check the publication of Yabuuchi et al. (J. Mater. Chem. A, 2021, 9, 15963–15967). Figure 3a in the Yabuuchi’s publication also have similar data with this publication, but they analyzed as the Ni migration to tetrahedral sites in Li layer.

There have been different interpretations regarding the degradation over cycling. Yabuuchi et al., did nice work in showing the Ni migration to the tetrahedral sites by refining the synchrotron XRD pattern of the over-delithiated NiO₂. In the absence of Li in Octahedral sites, Ni may migrate to the tetrahedral sites, but when Li is present in the Octahedral sites, Ni may not be able to migrate to tetrahedral sites (due to the high energy penalty). Fig. S16 (Fig. 17 in the revision) is just to show that after 100 cycles, the composite structure is maintained. The XRD pattern was taken on fully discharged (lithiated) CAM. The Work of Yabuuchi et al is about the structure of a CAM in its fully delithiated state, where Ni could migrate to the tetragonal sites. We think the two cases are not comparable.

Revisions:

- we have added the paper by Yabuuchi et al. (J. Mater. Chem. A, 2021, 9, 15963–15967) and added the discussion in the “Discussion” session, as below:

There have been different interpretations regarding the degradation over cycling. Yabuuchi et al., showed the Ni migration to the tetrahedral sites in the over-delithiated NiO₂. However, when Li is present in the Octahedral sites, Ni may not be able to migrate to tetrahedral sites due to the high energy penalty). Due to the different materials system and the cycling conditions, the degradation mechanism can be very different, but eventually the structural degradation leads to the formation of the electrochemically inactive rock-salts.

(5) Authors claims that “small anisotropic lattice expansion and contraction during cycling, they exhibit high structural stability”. However, the direct evidence was not provided.

Response: We believe the results in Figure S16 (Fig. S17 in the revision) on the high structural stability is strong evidence, say the consequence of the small anisotropic lattice expansion and contraction during cycling. In addition, we have included the following evidence:

i) In Fig. S4 (copied here as **Figure R6**), we showed the reduced amplitude of the H2/H3 transformation of the Li-deficient NM9505 (with 0.95Li) in the dQ/dV in comparison to the Li-stoichiometric one (NM9505 with 1.05Li). Such an amplitude reduction is the direct evidence of reduced anisotropic lattice expansion and contraction, as already demonstrated in one recent work by Y-K. Sun group (Ref.s H. Ryu, G. Park, C. S. Yoon, and Y-K. Sun, *J. Mater. Chem. A*, 2019, 7, 18580).

Figure R6. The dQ/dV curves of NM9505-0.95Li (red) and -1.05Li (green).

Y-K. Sun et al. published a very comprehensive paper (H. Ryu, G. Park, C. S. Yoon, and Y. Sun, *J. Mater. Chem. A*, 2019, 7, 18580). They used in situ XRD to characterize the lattice change during battery charging/discharging, and combined with the dQ/dV study. They found that these two methods correlated very well. When the c-axis change is large, the corresponding dQ/dV peak intensity at 4.2 V (H2-H3) is high. As shown in Figure S4 (Fig. S5 in the revised version), NM9505-1.05Li showed a sharp dQ/dV peaks, indicating a large lattice change. In comparison, the dual phase NM9505-0.95Li had a reduced peak intensity.

ii) In another relevant work on synthesis of Li-deficient $\text{Li}_{1-x}\text{NiO}_2$ (LNO) with the composite structure or dual phase (ref. 37 <https://doi.org/10.21203/rs.3.rs-1450650/v1>), we did in situ XRD for the Li-deficient composite LNO and stoichiometric layered LNO, showing that composite structure exhibited reduced c-lattice change compared to the normal layered phase (Figure 6).

Figure R7 (Figure 6 in the paper: <https://doi.org/10.21203/rs.3.rs-1450650/v1>). Structural evolution of the dual-phase 600C-6h cathode during the delithiation/lithiation processes compared to that of the single-phase LNO based on operando XRD examinations. Charge-discharge curves at 0.2 C of a, LNO and b, 600C-6h in the range between 4.1 and 4.8 V during the initial charge. i, bubble plot of the calculated lattice c in the crystal structures during the H2 to H3 phase transitions of the two cathode materials in the initial charge (the area of bubbles represents the Bragg peaks area).

Revisions: we made the following changes:

- i) added discussion/evidences on the reduced lattice change and added the new reference (H. Ryu, G. Park, C. S. Yoon, and Y. Sun, J. Mater. Chem. A, 2019, 7, 18580).
- added the reference to our early paper (<https://doi.org/10.21203/rs.3.rs-1450650/v1>)
- In addition, we revised the Abstract to weaken our statement:

Through multiscale-correlated experimental characterization and computational modeling, we unveil the crucial role of Li-stoichiometry in governing phase progression and crystallization during the calcination process. Specifically, Li-deficiency introduced in calcining NM9505 inhibits crystal growth and inter-particle fusion, consequently leading to small-sized layer-rocksalt composite, with the desired low anisotropic lattice expansion and contraction during charging/discharging.

6) Figure 5 f, g is not clearly described. I think that if the composite model was used similar crystallization process is expected for LiNi_{0.95}Mn_{0.05}O₂ and Li_{0.90} Ni_{0.95}Mn_{0.05}O₂.

Response: We have added more description to the Figure 5, f and g, to explain the different crystallization process between the Li-deficient and Li-stoichiometry cases.

Revisions:

1) we add description in the Discussion session as below:

When the Li source is just enough or below the stoichiometry ($Li/TM < 1.0$ or ~ 1.0) (see Fig. 5f), most Li is incorporated into the particles, facilitating lithiation-driven phase transformation and crystal growth, resulting in smaller primary particles with a composite structure, consisting of rocksalt and layered phases. When extra Li is added during calcination ($Li/TM > 1.0$) (see Fig. 5g), not all the LiOH reacts with the cathode precursors, and extra lithium salt exists in a molten state surrounding the primary particles. A liquid phase sintering mechanism gets activated due to the faster mass diffusion through this liquid phase. Therefore, small primary particles tend to merge into larger ones, similar to the synthesis of single-crystal NMC cathode materials via the molten salt method⁴⁶, which is sometime characterized as the Ostwald ripening process. Consequently, the overall primary particle size ends up being larger for Li-excess conditions.

2) We have also added description to the caption of Figure 5 (f, g):

f-g, Schematic representation of the sintering induced particle growth mechanism observed during the calcination of Ni-rich cathode primary particles with different amount of lithium salt. TM(OH)₂ precursors, LiOH salt particles, and lithiated LiTMO₂ particles are denoted by orange, green and blue, respectively. Scheme (f) indicates that, under lithium poor condition ($Li/TM < 1.0$), or stoichiometric amount ($Li/TM \approx 1.0$), all the lithium is consumed in the reaction with the cathode precursors, and no excess lithium salt exists. No fusion among individual grains occurs during calcination without excess Li salt. On the contrary, scheme (g) indicates that with the presence of excess lithium salt ($Li/TM > 1.0$), significant amount of the non-reacted molten lithium salt exists adjacent to the cathode primary particles. This excess lithium salt in liquid phase acts as a sintering aid and leads to substantial amount of sintering induced particle growth through inter-particle fusion.

Reviewer #2

In this manuscript, the authors investigated the impact of LiOH/M(OH)₂ molar ratio on the primary particle growth and the electrochemical performance of high-Ni, Co-free cathode materials. I think this is an interesting work done by a group of well-respected battery research scientists. However, some major changes and clarification are needed before this manuscript may be considered for publication at Nat. Commun.

(1) More electrochemical tests should be carried out to demonstrate the advantage of the Li-deficient Ni-rich cathode. What about the rate-performance, Li diffusivity, full cell performance? The authors only showed half-cell performance from one cell in each case. This is not convincing enough if this manuscript is going to be published at Nat. Commun. If we only compare the half-cell performance, I think the 1.022Li-sample looks very good. Then, what is the point of using Li-deficient Ni-rich cathode?

Response: We thank the reviewer for the suggestion of making more electrochemical tests. Indeed, we have tested their rate performance and diffusivity, as shown in **Figures R8 and R9**. The results show that the rate performance of Li-deficient NM9505 is worse than Li-excess ones, likely due to the high concentration of the RS component. Further optimization is needed, possibly through refining the Li-stoichiometry and phase constituent components.

Figure R8. Rate performance of the NM9505 with different Li stoichiometries.

Figure R9. Li diffusion coefficient of NM9505-0.95Li and 1.05Li.

We agreed with the reviewer that the “1.025Li-sample” shows very good performance in terms of the first-cycle Coulombic Efficiency (CE) and cycling stability, which is also comparable to literature reports. However, the cycling performance is worse than the Li-deficient one (0.95Li-NM9505), of 90 % vs 80 % for 100 cycles, as shown in Figures 3e and S13 – see also **Figure R10**, with the y-dimension expanded for better view of the difference.

Figure R10. (a) Specific capacity and (b) capacity retention vs cycling number plots. (c) Capacity retention after 100 cycles.

We'd like to point out that we made electrochemical tests from at least 3 cells and the results included in this manuscript are the averaged values over the 3 cells. See one example of the cycling performance from NM9505 with 0.95Li and 1.05Li in **Figure R11**, showing the high consistency of the cycling data from the 3 cells.

Figure R11. Cycling performance of NM9505 with 0.95Li and 1.05Li. The error bar is the standard deviation from three cells at the same test condition.

See also the updated **Figure S6**, showing the cycling performance of NM9505 and NM9010 at 0.5C, showing the averaged data from 3 cells and the standard deviation.

Revisions:

- Revised Figure S14, with the expanded y-axis, for better view of the performance difference between the samples with different Li contents.

(2) More structural characterizations may be needed. The authors claimed a minor Li-containing rock-salt phase ($\text{Li}_y\text{TM}_{1-y}\text{O}$) of about 19.6 mol%. This is a significant amount. Can it be observed via STEM-HAADF?

Response: This is a great suggestion. The 19.6 mol% of rock-salt phase was resolved by refinement of the synchrotron XRD data. We did observe the spatial distribution of RS across the secondary particles by 3D TXM mapping, as shown in **Figure R2** in response to Reviewer 1's comment #1 (see also Figure S4 in the supporting).

By following the reviewer's comments, we have also taken STEM-HAADF from the local region of the primary particles, showing the layered-RS composite (see the response to response to Comment 2 by the Reviewer 1). The newly taken STEM-HAADF has been included in Figures 2 (d) and **S4**.

(3) What is the exact value of y in $\text{Li}_y\text{TM}_{1-y}\text{O}$? When the authors carried out the Rietveld refinements, did they consider other possible "second phases?" Can it be a Li-deficient layered phase? If it is really a Li-containing, NiO-like rocksalt, wouldn't it negatively impact the Li transport and the rate performance? This is why I suggest GITT measurements in my first comment.

Response: The Li content (y) is ~ 0.3, with the value varying from sample to sample, as listed in Table S2 (copied here, highlighted in green).

We tried different structure models in the refinement (for the best fitting of diffraction data) and found that the two phases, Li-poor rock salt and Li-rich layered phase (with Li content above 0.9)

Yes, the reviewer is right that the presence of Li-containing RS negatively impacts Li transport and rate performance, as shown by the rate testing data and GITT data (**Figure R9**). Some more work is needed to further optimize the concentration of the RS through tuning the composition and synthesis conditions, which will be reported in the future work.

Table S2. Refinement parameters of NM9505 with different Li contents.

Li/TM _{add}	RS-mol	a(RS)	a-L	c-L	Li-slab	Ni-mix	L-domain size	c/a	Li Occupancy (RS)	Li/TM	Rwp
0.90	0.232	4.083	2.882	14.222	2.6675	0.008	45.98	4.934	0.358	0.916	2.35
0.95	0.196	4.082	2.881	14.219	2.6617	0.021	52.09	4.934	0.359	0.906	2.35
1.00	0.033	4.074	2.878	14.211	2.6371	0.035	97.53	4.936	0.24	0.917	2.06
1.025	0.037	4.070	2.876	14.205	2.6360	0.018	99.59	4.939	0.310	0.951	2.18
1.05	0	-	2.874	14.2015	2.6337	0.026	118.74	4.941	0	0.950	2.03
1.10	0	-	2.871	14.191	2.6336	0.022	138.18	4.943	0	0.956	1.91

(4) The sintering time appears to be quite long, 600 C for 12h and then 720 for 12h. What is the point of having the 600 C step? The oxygen flow rate should be reported.

Response: The sintering at 600 C is to ensure molten Li can be well diffused across the secondary particles. We used 1L/min O₂ flow rate.

Revisions:

We added the O₂ flow rate into the **Experimental section**.

(5) The particle size distribution is obtained from the SEM images of the secondary particle surface. What about the interior of these secondary particles? Can the authors carry out some ion-milling or FIB experiments to check the inside?

Response: This is a great comment. We did not do the FIB to cut the sample but took SEM images from the broken particles, with the interior exposed; see one typical example shown in **Figure R11**. The primary particles are plate-like, elongated along the radial direction of the secondary particles, making it hard to do the comparison between different samples.

(6) In the abstract, the authors write " the anisotropic lattice expansion and contraction is suppressed in Li-deficient NM9505 upon cycling". I did not see any in situ electrochemistry-XRD data to support this claim.

Response:

This is a great comment. We have provided the response to the similar comment from the 1st reviewer (Comment (5)). Although we did not do in situ XRD for the Li-deficient NM9505, we took in situ XRD from Li-deficient LiNiO₂, showing the reduced lattice expansion and contraction. We have also weakened our statement in the Abstract as below.

Revisions:

Through multiscale-correlated experimental characterization and computational modeling, we unveil the crucial role of Li-stoichiometry in governing phase progression and crystallization during the calcination process. Specifically, Li-deficiency introduced in calcining NM9505 inhibits crystal growth and inter-particle fusion, consequently leading to small-sized layer-rocksalt composites, with the desired low anisotropic lattice expansion and contraction during charging/discharging.

(7) Fine-tuning the composition and microstructure of the layered cathode materials seems an interesting new trend. How does the Li-deficient Ni-rich, Co-free cathode compare with

the slightly Li-rich nickel oxide cathode and the Li-rich, Ni-rich Li_{1.09}Ni_{0.85}Mo_{0.08}O₂ cathode?

Response:

We agreed with the reviewer that fine-tuning the composition and microstructure of the layered cathodes is an interesting area. As shown from our own data, the electrochemical performance of the Li-deficient NM9505 is better than the Li-rich NM9505 (as shown in Figure S14). We believe the Li-residuals and the large-sized primary particles are the main issues for the Li-rich systems.

There are other reports on Li-rich, Ni-rich systems, such as the one reported by Tarascon Group *Energy & Environmental Science*, 16(3), pp.1210-1222.), showing as high as 91% capacity retention for 100 cycles. It is an interesting work but the high cycling stability was achieved through Mo doping in order to form disordered rocksalt, which is less cost-efficient compared to the Li-stoichiometry control approach.

Revisions:

We added the new reference (Energy & Environmental Science, 16(3), pp.1210-1222) into the paper, and added discussion:

These findings demonstrate an efficient route to structural and morphological tuning via Li stoichiometry for stabilizing Co-free cathodes, with the desired cost-efficiency by eliminating the extra coating/doping steps^{38,39,56}

Reviewer #3

The present manuscript with title "stabilising cobalt-free cathodes with lithium-stoichiometry control for sustainable lithium-ion batteries" by Chen et al. explores the effects of Li stoichiometry in the structure and electrochemical properties of Co-free $\text{Li}_x\text{Ni}_{0.95}\text{Mn}_{0.05}\text{O}_2$ (NM9505) using synchrotron X-ray diffraction and multiscale modelling as main characterisation tools. Using as a proof of concept materials Li-deficient (0.95) NM9505 they show that Li deficiency in the structure is key to hindering crystallization and interparticle fusion during calcination, leading to the formation of composites containing intergrown layered and rocksalt phases which are responsible for good performance of this material with respect to the stoichiometric material in terms of ICE, capacity retention and negligent voltage fade. The work is relevant to the battery research and industry community and therefore, we would like to recommend the manuscript to be accepted after some major corrections.

(1) The composition of the RS phase observed in the Li-deficient samples is not entirely clear. What RS phase was used in the Rietveld refinement? More detail should be given about what the stoichiometry of this RS might be, even if this is tentative.

Response: Yes, the Li stoichiometry was obtained from refinement, which is about 0.3 but slightly varying in different samples, as given in Table S2 (copied here; highlighted in green)

Table S2. Refinement parameters of NM9505 with different Li contents.

Li/TM add	RS-mol	a(RS)	a-L	c-L	Li-slab	Ni-mix	L-domain size	c/a	Li Occupancy (RS)	Li/TM	Rwp
0.90	0.232	4.083	2.882	14.222	2.6675	0.008	45.98	4.934	0.358	0.916	2.35
0.95	0.196	4.082	2.881	14.219	2.6617	0.021	52.09	4.934	0.359	0.906	2.35
1.00	0.033	4.074	2.878	14.211	2.6371	0.035	97.53	4.936	0.24	0.917	2.06
1.025	0.037	4.070	2.876	14.205	2.6360	0.018	99.59	4.939	0.310	0.951	2.18
1.05	0	-	2.874	14.2015	2.6337	0.026	118.74	4.941	0	0.950	2.03
1.10	0	-	2.871	14.191	2.6336	0.022	138.18	4.943	0	0.956	1.91

(2) 3D TXM mapping for the Li stoichiometric phase should be provided for reference so one can understand how these data compare to the 3D TXM mapping provided for the Li-deficient phase. It should be clarified in the manuscript reason for having both Ni^{3+} and Ni^{2+} ions present in the Li-deficient material.

Response:

We have added 3D TXM mapping for the Li-stoichiometric one, as shown in **Figure R4**, showing more uniform distribution of Ni²⁺ across the secondary particles, compared to the Li-deficient one.

In the Li-stoichiometric phase, Ni is mostly in Ni³⁺ and only ~ 5% Ni²⁺ due to the 5% Mn. In the Li-deficient phase, the Ni²⁺ comes from the presence of the rocksalt phase, besides the 5% from the presence of the 5% Mn.

Revisions:

- We have added the TXM mapping for the 1.05Li for comparison in **Figure S4**.

(3) In line 121 the authors mention a slight overpotential in the Li-deficient NM9505 sample. It should be indicated in which potential window this is observed. This is explained with the presence of RS but no more explanations are provided. The authors should clarify this statement.

Response:

The overpotential can be clearly reflected in the dQ/dV curves as shown in Figure S5 (as copied below). At the voltage window 4.2-4.3 V, the H2 to H3 transition peak of NM9505-0.95Li shifted to a higher voltage.

Figure S5. The dQ/dV curve of NM9505-0.95Li and 1.05Li

Revision: we add explanation regarding the overpotential as below:

“The slightly higher overpotential at the end of charge and discharge in NM9505-0.95Li is probably due to the presence of RS that slowed Li diffusivity, which is also shown by the shift of the H2 to H3 transition peaks in the dQ/dV curve in Figure S5.”

(4) In line 125 (Figure S4), the authors should highlight the different H1, M, H2 and H3 phase transitions in the Figure.

Response:

We added labels, as in the last response, and also updated the Figure S4 (Fig. S5 in the revision) with labels.

(6) In line 154, the authors highlight that the initial capacity of NM9010 is 160 mAh g⁻¹. However, Figure S5 shows that this capacity is close to 200 mAh g⁻¹. There is a large decay in capacity after 3-4 cycles from this initial capacity to 160 mAh g⁻¹ which the authors have not mentioned. How many repetitions of the electrochemistry in this material have been made? Is this common in all repetitions? If so, an explanation should be provided as to why this is happening.

Response:

The initial capacity of NM9010 at 0.1C was 198 mAhg⁻¹, and the capacity at 0.5C was 160 mAhg⁻¹. We have corrected the values in the revised manuscript.

Fig. S6. Battery cycling performance of NM9505 and NM9010 at 0.5C.

We have at least 3 repeats for each condition. In the updated figures, we added the standard deviation for both NM9505 and NM9010. As shown in the Figure S6, all the cells showed the similar trend. NM9010 showed poor rate capability possibly due to the high Mn content while lack of the presence of Co in the structure.

Revision:

We updated the Figure S6, added with the standard deviation.

(7) Microscopy images of the different samples after long cycling should be provided to confirm that there is less microcracking on the Li-deficient NM9505.

Response:

We agreed with the reviewer that microscopy images may provide evidence on the reduced cracks in the Li-deficient NM9505. We did not take the microscopy images but took the synchrotron XRD of the cycled electrodes, which provide similar information on reduced

cracking, by showing the retention of the RS component in the Li-deficient NM9505 (Figure S17).

(8) In Figure S9. Li_2CO_3 peaks should be indicated in the Figure.

Response:

We add standard reference for Li_2CO_3 to the Figure S9, as copied below. In the revised manuscript, the Figure S9 was updated.

Figure S9 (rev). Zoom-in view of synchrotron XRD patterns to show the presence of Li_2CO_3 with excess Li.

(10) In Figure S11, references for the different oxidation states of Ni and Mn should be provided to allow a better comparison between these and the data provided in the manuscript.

Response:

We added reference to in Figure S11 (Fig. S12 in revision), see the **Figure R2** in the response to the first review's comments (# 2).

(11) Some typos were found in the manuscript: e.g. Line 72 "Materials", Line 144 "valance", Line 194 "stoichiometry..Even"

Response:

We have corrected all the typos and made thorough proofreading of the manuscript.

REVIEWERS' COMMENTS

Reviewer #1 (Remarks to the Author):

Please see my comments in the attachment.

Reviewer #2 (Remarks to the Author):

The authors have addressed my comments. I believe it may be considered for publication.